# Evaluation of the Performance of Gabion Walls as a High-Energy Rockfall Protection System Using 3D Numerical Analysis: A Case Study

Zekai Angın and Olguhan Şevket Karahasan *

Department of Civil Engineering, Karadeniz Technical University, 61080 Trabzon, Turkey; angin@ktu.edu.tr
* Correspondence: olguhan@ktu.edu.tr

**Abstract:** Cases of rockfalls that occurred on a slope in the Selendi District of Manisa Province are evaluated in this article. Field studies are evaluated and different measures are examined to reduce rockfall risk. Drone flights are used to evaluate previous studies and obtain 1/1000 scale digital topographic maps. These maps are used to create a 3D (three-dimensional) solid model of the project site, and on-site surveys are conducted to identify source rock locations and free blocks that pose a risk. Those 3D analyses are used to determine the paths, jump heights, and energies of the blocks in motion. The data from the 3D maps are used to determine the most appropriate remediation methods. The structural behavior of the recommended gabion wall, which is designed at a certain height and width as a result of rockfalls, has been examined. Structural behavior is determined by simulation based on the finite element model. Within the scope of this study, the ANSYS Workbench program is used. The "Explicit Dynamics" analysis type in ANSYS Workbench was chosen to examine the rockfall effect.

**Keywords:** rockfall; finite element model; ANSYS; structural behavior; gabion wall; 3D analysis





## 1. Introduction

Climate change is a result of human effects and natural events that occur with the deterioration of the gas balance in the atmosphere and whose results are observed worldwide. These changes disrupt the order of climates and adversely affect human life. Especially in recent years, the effects of climate changes have become more evident throughout the world. Various effects of climate change are observed in Turkey [1]. Rockfall is one of the events caused by climate change. Such events that cause loss of life and property are seen in a large part of Turkey. Eastern Anatolia, Eastern Black Sea, and Western Black Sea are the first places in terms of the number of rockfall events and the number of people affected by the event. Mass movements sometimes cause loss of life and property [2–6]. Destruction of vegetation, damage to cities and roads, and covering of agricultural areas by inefficient materials are other negative results of mass movements.

Rockfall is the most common natural phenomenon in mountainous regions. Rockfall events usually affect a small area. They occur at variable frequency intervals, happening suddenly with high kinetic energy. These reasons make it difficult to take immediate action when a rockfall event occurs. Blocks ranging from a few kilograms to tens of tons each year cause traffic problems, damage to structures and vehicles, and sometimes deaths. It is called a natural hazard that an individual and superficial rock in a rock slope loses its stability due to weathering, internal and external factors, and falling under the influence of gravity. Considering all these, it is necessary to assess the risk of rockfall and take necessary measures before the event occurs in order to protect endangered residential areas and infrastructure [7]. Depending on the slope of the topography during rockfalls, movement types such as free falling, jumping, and rolling can be observed. As a result of the falling

rock blocks losing their energy, there may also be block slides along with the rolling motion of the block.

The purpose of the techniques applied to prevent rockfall events is not to increase the safety factor, but rather to prevent and/or minimize the disasters that may arise from rockfalls. Various methods are used to control and protect against rockfalls. These methods can be grouped under three main headings as wire mesh systems, barriers, and holders. Many studies have been conducted on rockfalls and their precautions. An experimental study by [8] utilized a full-scale pendulum to test the impact resistance of a gabion-cushioned concrete barrier, revealing significant reductions in both contact force and deflection demand, by 95% and 70%, respectively. Ref. [9] introduces a criterion for assessing the durability of protective embankments against rockfall impacts, derived from comprehensive real-scale impact studies on diverse embankment types. This method correlates the deformation of the embankment's downhill side with the kinetic energy of the impacting rock. Developed through a rigorous scientific method, this criterion was applied and validated on 98 embankments across France and Switzerland, demonstrating its effectiveness and reliability in determining the structural integrity of embankments against rockfall threats. Ref. [10] introduced an innovative method to assess wire meshes' durability against rock impacts, which is key in safeguarding infrastructure with catch fences. The study found longer meshes to be more effective in absorbing impact through deformation, with supporting cables reducing lateral compression. This research aids in refining catch fence designs by understanding wire mesh behavior under stress. Ref. [11]'s research showed that incorporating polyethylene (PE) fibers and expandable polyethylene (EPE) foams into rock-shed constructions significantly boosts their ability to withstand impacts, outperforming traditional materials by offering better energy absorption and durability. Another study utilized DEM simulations to show that during rockfalls, the energy absorption of embankments depends on their geometric designs, such as crest thickness and slope inclinations, rather than material stiffness. It suggests thicker crests for efficient space use and comparable energy dissipation [12]. Ref. [13] found horizontal reinforcement more effective than vertical reinforcement in reducing deformation and preventing failures in soil embankments, offering guidance for optimizing rockfall protection designs. Ref. [14] evaluated the performance of a rockfall ditch in the prevention of rockfalls by three-dimensional rockfall analysis. Three-dimensional digital surface models of the study area were created by researchers using drones. Three-dimensional rockfall analyses were performed using RocPro3D (version 6.2) [15] three-dimensional rockfall analysis software. The paths, jump heights, and energies of the blocks in motion were determined. According to the 3D rockfall analysis, some parts of the rockfall ditch constructed at the upper elevations of the study area are insufficient to prevent rockfalls. Ref. [16] examined the rockfall hazard by two-dimensional rockfall analyses. In the rockfall analysis, the run-out distance, jump height, kinetic energy, and velocities of the blocks were determined. The results obtained from the rockfall analysis were used to map the areas of possible rockfall hazard zones. It was seen that due to the topographic, atmospheric, and lithological features of the study area, the precautions in the literature are insufficient. For this reason, the researchers suggested that the residents should be evacuated from the danger zones first and then the hanging blocks should be cleaned in reachable locations by taking safety precautions. Ref. [17] investigated the rockfall hazard around Afyon Castle (Turkey) by two-dimensional rockfall analysis. Fall distance, bounce height, kinetic energy of rocks, and velocity profile were simulated. They suggested installing a protective fence in the study area against the danger of rockfall. Ref. [18] obtained maps of the study area (Kilis in Turkey) using geographical information system techniques and remote sensing data to monitor possible rockfall risks. They overlapped the maps using geographical information systems and obtained a rockfall risk map. Ref. [19] assessed the rockfall hazard near Ankara Castle (Turkey) in order to propose appropriate solutions. Then, they presented precautions based on the analysis results and assessed the risk based on a rockfall hazard rating system. Two-dimensional rockfall analyses were performed to assess the danger area around the Citadel. Since

there was a residential area nearby, they did not consider it possible to build ditches without disturbing the environment. Moreover, it was seen that rock bolt assembly was also impractical for the case. The researchers suggested catch barriers to protect settlements from rockfall events. Various scenarios were simulated to investigate the effectiveness of catch barriers in the event of a rockfall. In the worst-case scenario, they saw that catch barriers protected settlements from damage in the event of a rockfall. Ref. [20] investigated the geological characteristics of a study area in Zonguldak, Turkey. Laboratory and field studies were applied. Rockfall analyses were applied using computer software at the same time. They have stated that the capacity of the barrier is the amount of energy required to break the barrier by the falling rock. As a result of the analyses, it was decided to use a barrier of optimum distance and height to protect the road from rockfall. Ref. [5] aimed to examine the measures taken to mitigate rockfall hazards in the Sumela Monastery complex. The study involved several stages, including field studies, probabilistic rockfall analysis, finite element simulations, and cleaning and controlled removal of risky rock blocks. The field studies were conducted to evaluate the regional geology and identify the locations, dimensions, and risk level of the risky rock blocks. The probabilistic analysis was used to calculate the maximum run-out distance, velocity, kinetic energy, and bounce heights of falling blocks. The finite element simulations were conducted to evaluate the damage status on the vaulted building, and the cleaning and controlled removal method of the risky blocks was determined. The study concluded by the elimination of the rockfall hazard, and the monastery complex was opened for visitors in 2021. Rockfalls are a significant geological hazard in sloped and mountainous areas, posing serious threats to both human life and infrastructure. Research in this field discusses the dynamics of rockfall events and explores various methods that can be used to mitigate the risks caused by these events. Among these methods, gabion walls stand out as a noteworthy solution. Gabion walls can be described as flexible structures consisting of containers filled with stone or similar hard materials, enclosed by galvanized wire mesh. In the literature, these walls are often studied for their ability to slow the speed of falling rocks and absorb their energy. Additionally, they are favored for their environmental friendliness and aesthetic appeal. A review of the literature reveals numerous studies evaluating the effectiveness of gabion walls against rockfalls. These studies analyze different factors such as the structural properties of the walls, material choices, dimensions, and application methods, detailing how these walls reduce the risk of rockfalls. Moreover, the structural performance of gabion walls has also been examined analytically and experimentally by many researchers. Particularly, the flexible nature, high energy absorption capacity, and ecological compatibility of gabion walls are frequently emphasized in these studies [8,21–23]. When the literature is examined, it is clearly seen that rockfall events are frequently studied and solutions are investigated. In these studies, soil/rock characterization was evaluated, and field and numerical studies were carried out. The precautions suggested to be taken in these studies have not been examined for structural integrity. However, examining structural integrity provides a more accurate and realistic approach to the situations that may be encountered in the field. Unlike recent studies in the literature, especially in Turkey, within the scope of this study, the structural behavior of the gabion wall, which is designed at a certain height and width as a result of rockfalls, has been examined. Structural behavior was determined by simulation based on the finite element model.

Selendi Stream Basin, which is also included in the study area, is one of the sub-basins in the Gediz Basin which is located in West Anatolia. It is located approximately between 38°39′41″–39°00′47″ north latitudes and 28°39′14″–29°11′41″ east longitudes (Figure 1). The study site, located in the Selendi Stream Basin, is located in Çortak Village, Selendi district of Manisa province. Severe erosion is observed in the study area due to lithological and geomorphological properties. The high drainage density, the prevalence of badlands topography resulting from surface erosion, gully erosion areas that appear in many different forms, and the sharp ridges separating them from each other, can be considered as leading evidence of severe erosion. This process, which especially affects rocks with low strength,

also increases rockfall events in rock masses located at higher elevations. In this article, (a) field studies were evaluated and different measures were examined to reduce rockfall risk. (b) Drone flights were used to evaluate previous studies and obtained 1/1000 scale digital topographic maps. (c) These maps were used to create a 3D solid model of the project site, and on-site surveys were conducted to identify source rock locations and free blocks that pose a risk. (d) 3D analyses were used to determine the paths, jump heights, and energies of the blocks in motion. (e) The data from the 3D maps were used to determine the most appropriate remediation methods. (f) The structural behavior of the gabion wall, which is designed at a certain height and width as a result of rockfall, has been examined. Structural behavior was determined by simulation based on the finite element model.

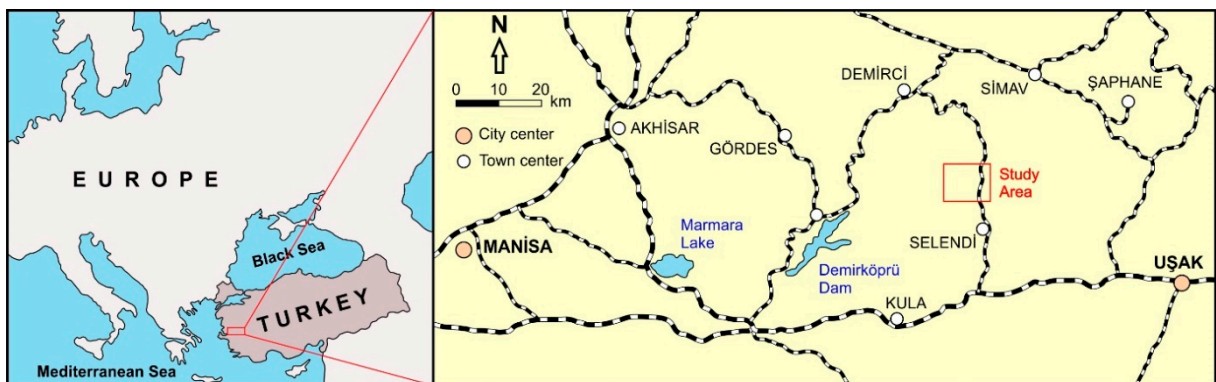

**Figure 1.** Location map of the study area.

## 2. Geology of the Study Area

Selendi basin is located between Demirci basin in the west and Uşak-Güre basin in the east. There are Simav and Gediz grabens in the north and south of the basin, respectively [24]. Dikendere Volcanites, which consists of rhyolite, rhyodacite, and tuff, outcrop in the study area (Figure 2).

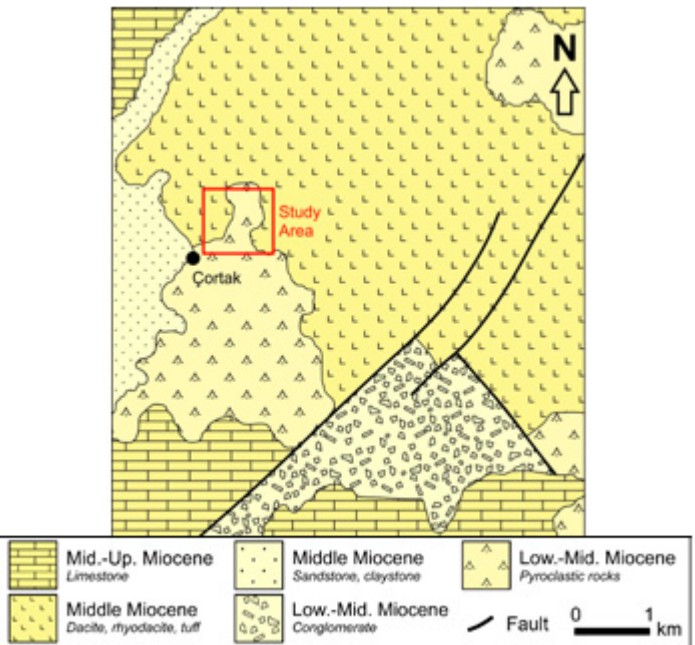

**Figure 2.** Geological map of the study site.

The name of the unit formed in the first volcanic phase in the Tertiary period of the region was given by [24]. Argillisation is common in the units. The upper contact zone

of the unit consists of Quaternary aged alluvium and slope debris. The age of the unit is given as Middle Miocene. There are rhyolitic and dacitic rock masses, especially at higher elevations of more than 800 m, in and around the study site. At lower elevations, low-strength tuffs outcrop.

## 3. Field Studies and Measurements

### 3.1. Rockfall Potential of the Study Area

Investigation of rock slope stability is essential for designing many engineering processes such as open cast mines, highways, and natural slopes. Slope designs made using appropriate and correct methods not only increase the stability of the slope, but also provide the opportunity to work in a safe environment by reducing accidents. Failures in rock slopes are mostly due to discontinuities in the mass. Most rock slope problems depend on geometric relationships between discontinuities. Therefore, kinematic evaluation of discontinuities is an important issue in the field of rock engineering.

Slope stability is generally evaluated with rock mass classification systems such as kinematic analysis, limit equilibrium analysis, numerical analysis, and Slope Mass Rating (SMR) [25–27]. In kinematic analysis, internal friction angle is used as data from slope geometry and shear strength parameters of discontinuities. These analyses can be used where failures in rock slopes are controlled by discontinuities [28–30]. Possible failure types (planar, wedge type, and tipping type failures) in rock slopes can be determined by kinematic analysis.

If any danger of failure occurs as a result of the kinematic analysis, the possible danger is investigated with limit equilibrium analysis. Limit equilibrium analyses consider shear strength along the failure plane, pore water pressure, and external forces such as maximum horizontal ground acceleration.

Rock mass classifications have been used successfully for years in tunnel and underground mining [31,32]. Some rock mass classifications developed for underground excavations were applied to slopes in the following years [32] or rearranged [33,34]. In this study, the studied slope was evaluated using the classification system suggested by [35]. While making the classification, the data obtained from field measurements and laboratory experiments were used (Table 1). This value shows that the rock slope in the study area is classified as "medium risk slope (risk class III)". When Table 1 is evaluated, it can be seen that it may be necessary to take easy protection (for example, bolts, meshes, removing unstable blocks, simple lightweight fences) in order to minimize the risks that may arise from rockfall events in the study site. However, it has been seen that as a result of the detailed field studies carried out in the study area, the dimensions of the moving blocks have reached 2 m, and these blocks have reached areas very close to the village by moving approximately 400–500 m from the source rock (Figure 3). Although it was desired to prevent rockfalls by digging trenches (width 2 m, depth 1 m) on the slope, these trenches were effective only for 50 cm blocks (Figure 4). For this reason, trenching on the slope is not an effective solution. Although enlargement of the ditches is considered, it is thought that these ditches will not be sufficient to stop the movement of the blocks reaching 2 m.

**Table 1.** Classification of slopes according to rockfall risk and the situation of the rock slope, which is the subject of this study, within this classification system.

| Risk Class | Total Weighted Score 1–100 | Risk | Indicative Protection Measures (the Choice is Site Specific) |
|---|---|---|---|
| I | <20 | Very Low | Not necessary. May be sparse spot interventions. |
| II | 21–40 | Low | In limited extent |

**Table 1.** *Cont.*

| Risk Class | Total Weighted Score 1–100 | Risk | Indicative Protection Measures (the Choice is Site Specific) |
|---|---|---|---|
| III | 41–60 | Medium | Light measures (such as bolts, nets, removal of unstable blocks, simple light fences) |
| IV | 61–80 | High | Combination of active (such as bolts, anchors) and passive (such as nets, wire rope cables, buttress walls, fences removal of unstable blocks) measures |
| V | 81–100 | Very High | Critical state of stability, combination of generalized or/and strong active and passive measures. Residual risk to be accepted. |

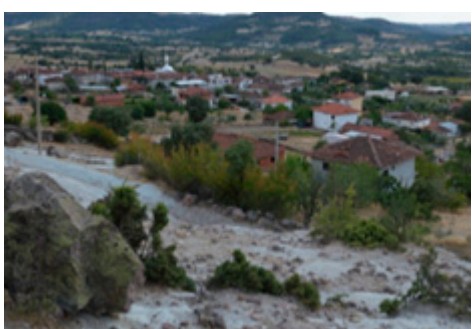 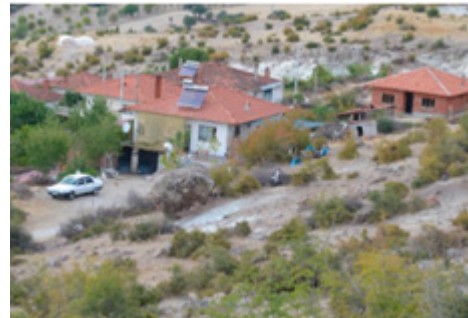

**Figure 3.** Large blocks reaching several meters in size (5–10 tons) breaking off from the source rock and reaching areas very close to the village.

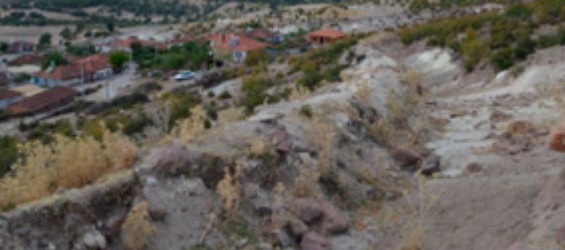

**Figure 4.** Trenches dug to reduce the risk of rockfall; their width reaches 2 m and depths up to 1 m.

Gabion walls can be effectively used as a method due to their many advantages, especially if there is a suitable area. Gabion walls are generally designed to eliminate the risks that may arise from large blocks with low bounce height but very high energy. This situation is also present in the study area. In addition, the fact that there is a buffer (flat) zone between the place area and the base of the slope shows that structures such as gabions can be applied on the slope.

Seismicity of a region defines the geographical distribution, occurrence times, magnitudes, mechanisms, and damage of earthquakes that occurred in that region. Seismicity studies are mostly carried out to evaluate the seismic risk of a region. In order to compare a potential earthquake effect and the blast effect to be applied in the scope of this study, the necessary data was obtained from the Turkey Earthquake Hazard Map, taking into account the Turkey Building Earthquake Regulation 2018 [36] (Figure 5).

In this case, considering the 2018 TBER; "earthquake ground motion level" is taken as earthquake level (EL-2), exceedance in 50 years is 10% (recurrence period is 475 years), and "local soil class" is ZC (weak rocks with weathered cracks). The following values are obtained.

Ss (Short period map spectral acceleration coefficient) = 0.646
S1 (Map spectral acceleration coefficient for 1 s period) = 0.163

SDS (Short period design spectral acceleration coefficient) = 0.802
SD1 (Design spectral acceleration coefficient for 1 s period) = 0.245
PGA (Peak ground acceleration) = 0.271 g
PGV (Peak ground velocity) = 15.235 cm/s

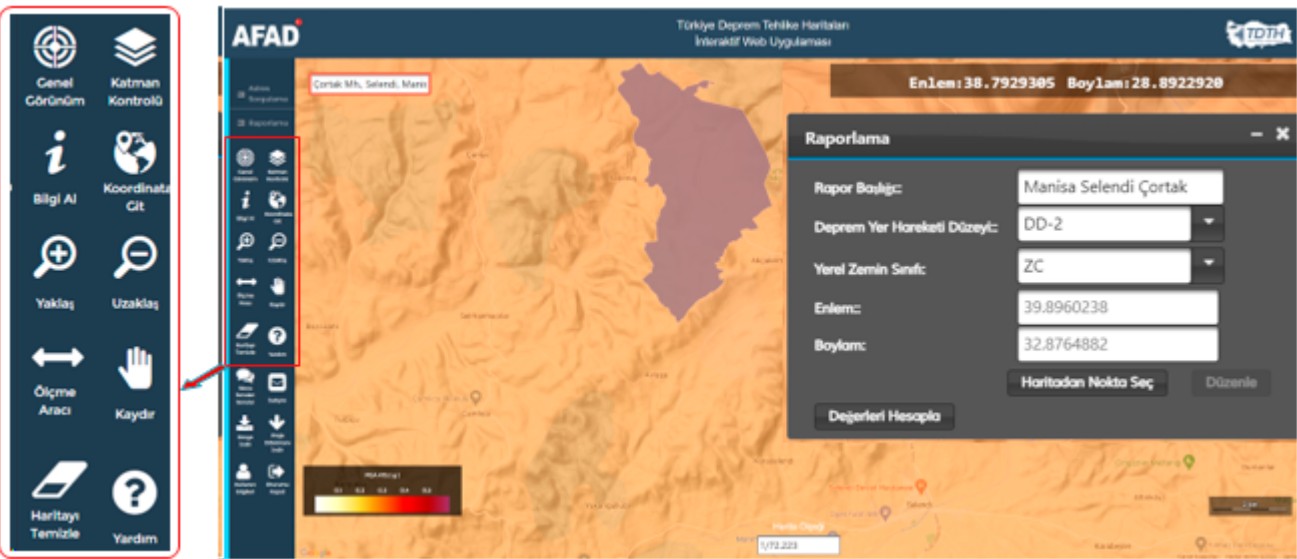

**Figure 5.** The maximum ground acceleration value according to the location of the study site in the AFAD Turkey Earthquake Hazard Map and the earthquake ground motion level with a probability of exceeding 10% in 50 years (recurrence period 475 years).

Research pits and their locations vary depending on the type, size, importance of the structure to be built, and the characteristics of the soil. A research pit was dug to a depth of 1.5 m and in five different locations. It was designed to reveal the soil profile, to determine the soil model to be used in the 3D rockfall analysis, and to determine the gabion wall foundation soil properties (Figures 6 and 7). After the lithological identification made in all research pits, talus of 50–70 cm was cut first and then tuffs with acidic character were cut. For this reason, it is recommended that approximately 50 cm of cover layer should be excavated and stripped before the gabion wall is built. Also, it is recommended that a foundation excavation be made so that the gabion base is at least 50 cm inside the tubes.

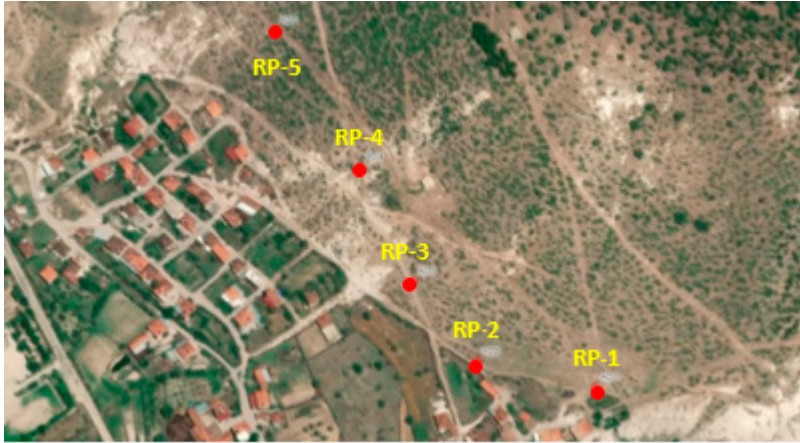

**Figure 6.** Location of the research pits opened within the scope of this study.

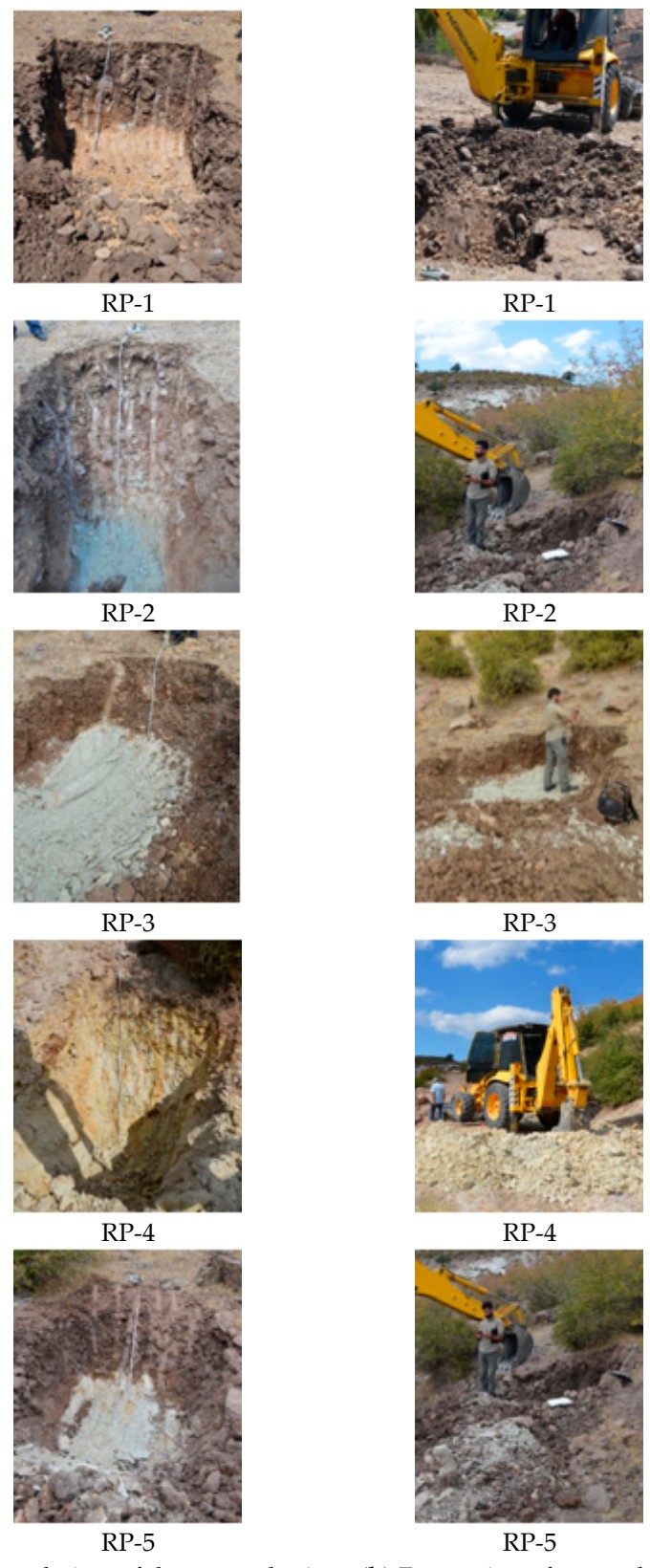

(**a**) General view of the research pits    (**b**) Excavation of research pits

**Figure 7.** Appearance of the research pits.

### 3.2. Creating a Three-Dimensional Model

The 3D rockfall analyses applied within the scope of this study were examined and evaluated under three main headings: (i) creating a three-dimensional terrain model, (ii) modeling the worst-case condition (fall analysis for the block with a mass of 10 tons), and (iii) evaluating the performance of the proposed rehabilitation design.

Çortak Village was established on the foot of a moderately natural dip slope. Dacidic rock masses presenting a nearly vertical topography on the upper floors of the slope have acquired a highly fractured structure depending on discontinuities and the degree of weathering. Blocks broken off from these source masses due to erosion and freeze-thaw processes move down the slope and cause rockfall problems. As a result of field observations and measurements, it has been seen that the dimensions of these blocks can reach 2.0 m in places. Within the scope of this study, it was thought that the most effective solution in order to minimize the risk of rockfall is a gabion wall. For this reason, 3D rockfall analyses were carried out in order to prevent the risk of rockfalls in the examined areas. Also, jump heights and energy amounts were calculated. Based on the results obtained, gabion wall positions and dimensions were determined. The 3D numeric surface model of the study area was produced as a result of digitizing the maps produced by drone flights in the study area (Figure 8).

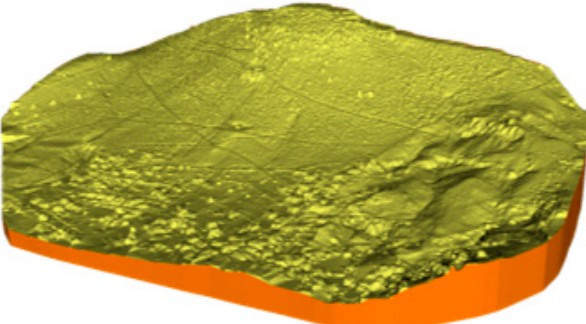

**Figure 8.** 3D solid model of the study site created using RocPro3D software (version 6.2).

A DJI Matrice 300 drone was used in photogrammetric studies. While using the DJI Matrice 300, measurements were made with centimeter precision with the D RTK 2 module as a ground station. By performing a two-way and three-turn terrain oblique flight with the Zennume L1 Lidar Camera, the tree cover was removed, and the surface model was created. Since the slope of the land is high, the resolution was prevented from decreasing at the lower elevations of the land by terrain flying (following the land surface) from an altitude of 100 m. The DJI Matrice 300 Drone was connected to a geofixed point using the DRTK2 Mobile station to use the RTK feature at the centimeter level. The data was processed as UTM WGS84 six degrees and a high resolution orthophoto was created. The triangulated irregular network (TIN) obtained from the digitized map and underlying the 3D rockfall analyses is given in Figure 9 and the digital elevation model (DEM) is given in Figure 10.

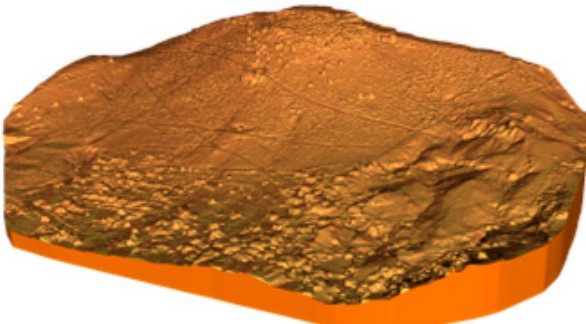

**Figure 9.** Triangulated irregular network (TIN) mesh created in RocPro3D software (version 6.2).

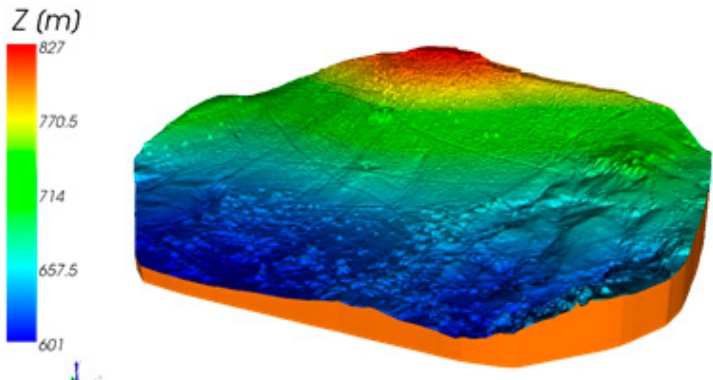

**Figure 10.** Digital elevation model created in RocPro3D software (version 6.2).

## 4. Results

### 4.1. Rockfall Analysis

Three-dimensional rockfall analyses were performed using RocPro3D (RocPro3D, 2014) three-dimensional rockfall analysis software. RocPro3D is a professional 3D stochastic trajectometric simulation software. This approach (Monte-Carlo type) offers the choice between Gaussian or equiprobable variables for the properties of soils and blocks (initial starting conditions, initial position, mass, rebound perturbations).

Thanks to this software, data such as the energy, jump height, etc., of a rock block that starts moving at a certain speed and has a certain mass at a certain point are obtained. Impact energy transfer and dissipation are calculated directly by this program and the results are presented visually.

Within the scope of this study, both the blocks that were found around the source rock and the blocks spread on the slope were examined, and the block geometry, maximum block size, and spreading areas were determined. As a result of field observations, it was understood that the block shape is generally spherical and that these blocks reached to the base of the slope. The most important reason for this situation is the weathering pattern seen in volcanic rocks. The exfoliation structures seen in the rocks due to weathering caused the rocks to take a spherical shape.

In the rockfall software used in this study, the block shape cannot be defined. The only input parameters are the position, mass, and initial velocity of the block. The output parameters are velocity, height, and path. For this reason, it is thought that the software used fully reflects the field conditions selected within this study.

During field studies, the locations of the blocks that traveled the longest distance and the shapes of these blocks were determined. It was observed that the angular blocks did not reach the gabion wall design line and the round blocks traveled the furthest distance. Considering this situation, a rockfall analysis model was established in the study. Such a scenario has not been encountered in the field. Because the distance between the source rock and the planned gabion is long, the blocks move separately at that distance, even if there is more than one rockfall event.

Blocks with a mass of 10 tons were dropped along a line from the highest point of the source rock, taking into account the worst-case scenario in the three-dimensional rockfall analyses. Fifty blocks were modeled in each region. In RocPro3D software, different geological units can be defined on the three-dimensional numeric surface model to model the rockfall and different return coefficients (Rn and Rt), and friction coefficients can be assigned for these units. In the study area, a single geological unit has been distinguished based on field observations in the region where the rockfalls occurred. The normal coefficient of restitution (Rn) was accepted as 0.75, the coefficient of tangential restitution (Rt) as 0.75, and the coefficient of friction (k) as 0.4. Cover thickness (less than 50 cm) and dacites outcropping in many areas, as well as the literature data, were taken into account. Rolling lines, jump heights, velocities, and energy values obtained from three-dimensional rockfall

analyses are given in Figures 11 and 12. When the analyses were examined, it was seen that the jump heights for a 10-ton block exceeded 2 m on the steep cliffs at the top, 1 m on average along the slope, and only turned into a rolling shape at the lowest elevations (the areas where the gabion design is planned). However, jump heights of 3 m were recorded in some parts of the lower elevations. At these points, the energy amounts vary between 4000–8000 kJ for a 10-ton block, and the average speed reaches 30 m/s. All risks that may arise from free blocks on the slope are blocked by the farthest gabion design from the source rock. However, in this case, the most effective solution is a gabion wall, as the distance will increase. The gabion wall should be applied along a line with as low of a slope as possible. It can be very difficult to secure the stability of the gabion wall in areas with high slopes.

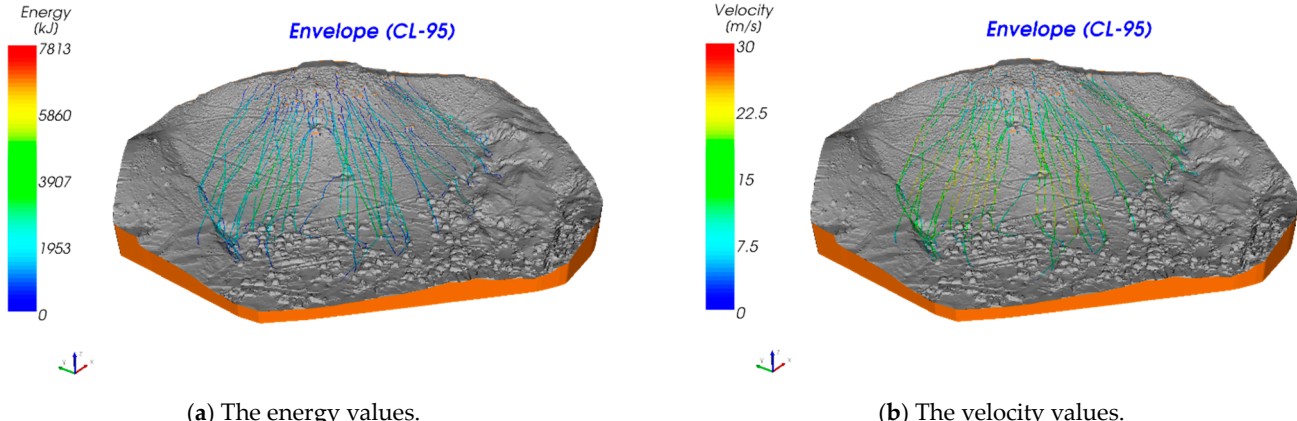

(**a**) The energy values.    (**b**) The velocity values.

**Figure 11.** The energy and velocity values determined for the falling/rolling of a 10-ton rock block over the source rock masses.

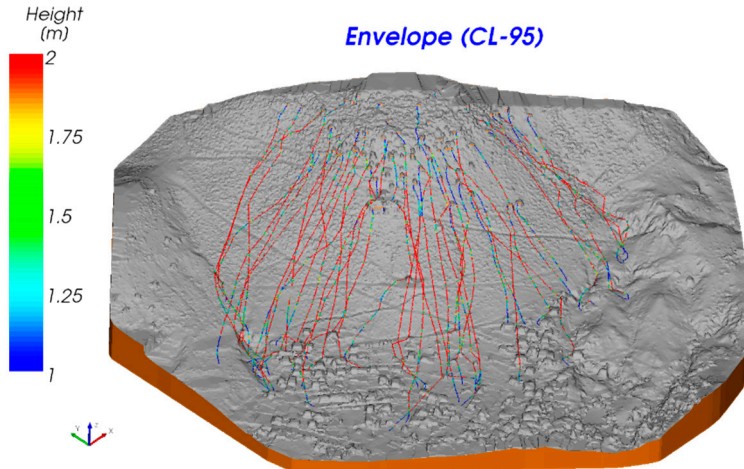

**Figure 12.** The jump height values determined for the falling/rolling of a 10-ton rock block over the source rock masses.

Gabion wall details are given in Figures 13 and 14. In Figure 15, the results of the 3D analyses made after the gabion wall design are given. When all analyses are examined, it is seen that the design is safe. The start/end coordinates of the proposed gabion walls are given in Table 2.

**Table 2.** Start/end coordinates of the proposed gabion walls.

|  | Start |  | End |  |
|---|---|---|---|---|
| Gabion-1 | 656,692.882 | 4,300,284.94 | 656,762.246 | 4,300,297.08 |
| Gabion-2 | 656,686.813 | 4,300,261.53 | 656,536.814 | 4,300,307.48 |
| Gabion-3 | 656,555.022 | 4,300,286.67 | 656,357.336 | 4,300,428.87 |
| Gabion-4 | 656,232.481 | 4,300,568.46 | 656,387.682 | 4,300,437.54 |
| Gabion-5 | 656,229.013 | 4,300,687.25 | 656,330.457 | 4,300,591.87 |

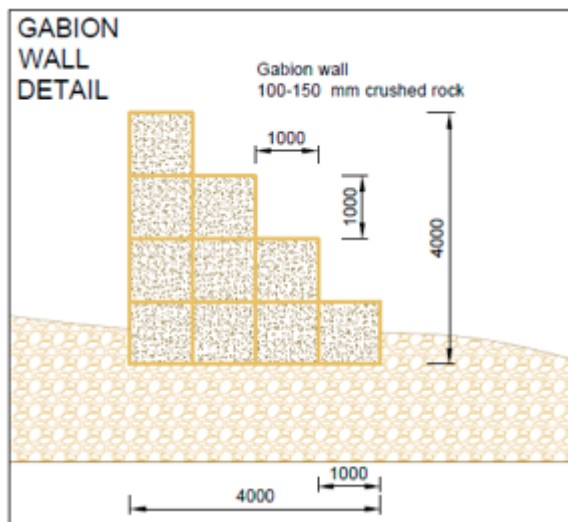

**Figure 13.** Suggested gabion wall detail.

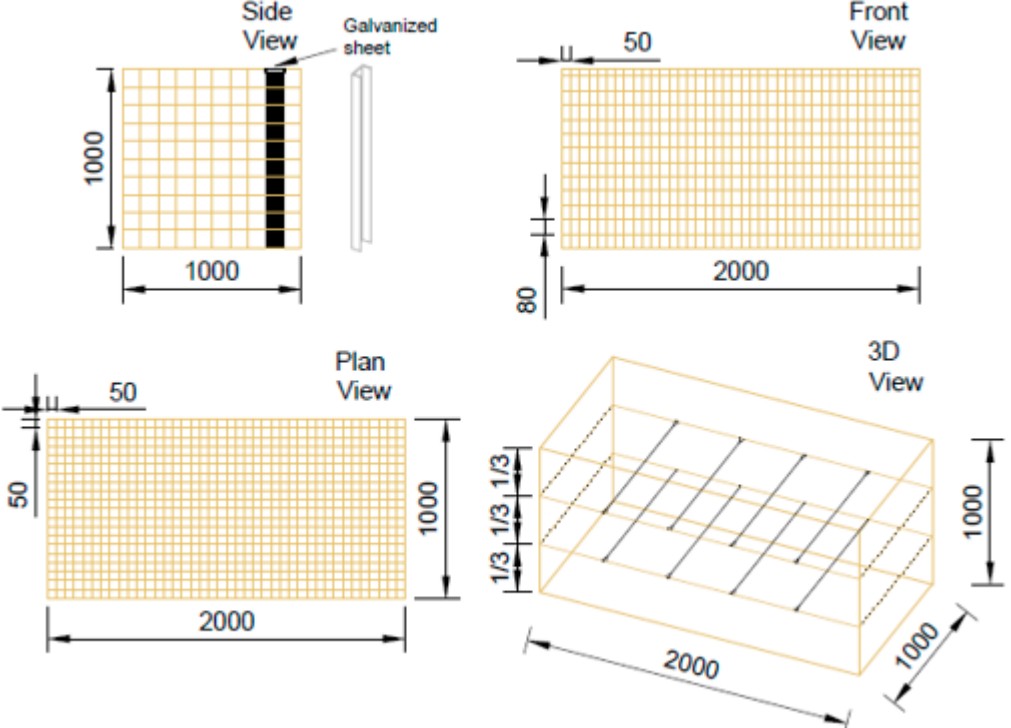

**Figure 14.** System dimensions of the gabion wall, view details from different directions, and pore sizes (in mm).

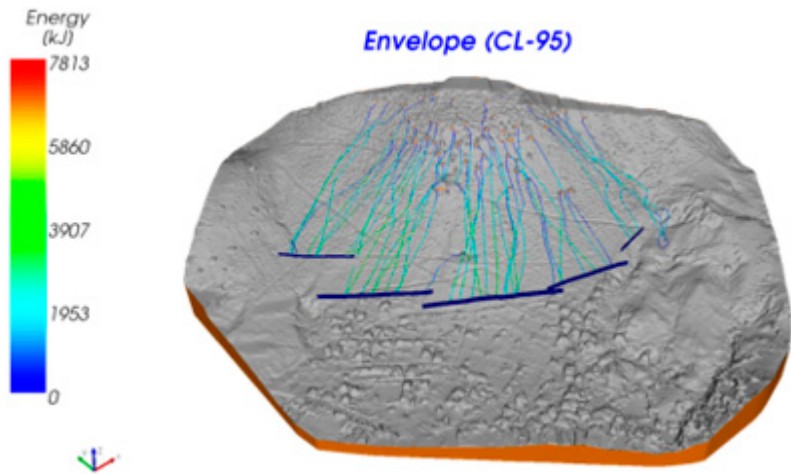

**Figure 15.** Three-dimensional rockfall analyses in case of designing a gabion wall.

*4.2. Performance Analysis for Gabion Wall*

After determining the appropriate location in rockfall studies, three different analyses should be made and evaluated for the gabion to be built.

Outside stability failure analysis (foundation bearing, etc.).

Inside stability failure analysis.

Integrity analysis against the impact.

4.2.1. Outside Stability Failure

External stability failures in reinforced earth structures.

Base slip control.

Eccentric limit and eccentricity control.

Assessment of the foundation bearing capacity.

Settlement estimate.

It can be checked as above.

The gabion wall structure, which is planned to be built in order to control rockfalls within the scope of this study, is not expected to carry lateral loads. Since there is no lateral load, base slip control, eccentric limit control, and eccentricity control are not possible.

In such structures, external stability failures are generally seen in the form of foundation settlement and bearing capacity failure. There are two types of failure caused by bearing capacity. The first is general shear (slip) failure, and the second is local shear (slip) failure. Local slip occurs when soft or loose soil units are present in the reinforced earth wall foundation. Bearing capacity calculations must be made to take into account both the strength limit state and the service limit state calculation. External stability analyses include the determination of the surcharge loads of the reinforcement earth system and the determination of settlements that may occur in the subgrade under its own weight.

Many empirical approaches have been developed to estimate the bearing capacity of discontinuous rock masses. In these methods, the RQD (Rock-Quality Designation) value (which is one of the parameters that best reflects the discontinuity properties), the uniaxial compressive strength value of the rock material, and the rock mass constants obtained from the Hoek-Brown failure criterion are used.

The approach developed by Bishnoi [37] is the most preferred in practical applications and is therefore still valid today. However, there are some limitations to this method. The method should be used in rock masses where the discontinuity spacing is greater than 0.3 m and the span is less than 10 mm (it can be up to 15 mm if the discontinuity is filled) while the foundation width is greater than 0.3 m.

$$qa = \sigma_{ci} \times N_j \tag{1}$$

$$N_j = Ks = \frac{3 + (S/B)}{10\sqrt{1 + 300(\delta/B)}} \tag{2}$$

qa: Allowable bearing capacity (MPa)
σci: Uniaxial compressive strength of rock material
Nj: The empirical coefficient depending on the discontinuity spacing
δ: Discontinuity span
S: Discontinuity spacing
B: Foundation width
Note: The following Table 3 is frequently used in practice when calculating Nj.

**Table 3.** Nj values corresponding to the spacing of discontinuity.

| Discontinuity Spacing (m) | Nj (or Ks) |
|---|---|
| >3.0 | 0.40 |
| 3.0–0.9 | 0.25 |
| 0.9–0.3 | 0.1 |

Ref. [38] suggested using the RQD values of the rock mass in the bearing capacity calculations. However, in these approaches, it is assumed that the bearing capacity value of the rock mass does not exceed the uniaxial compressive strength value of the rock mass. While calculating the bearing capacity, the average RQD value of the depth from the foundation base to the foundation width (B) is taken. Ref. [38] proposed equations using the uniaxial compressive strength (σci) values of the rock material together with the RQD value to estimate the bearing capacity of rock masses. Since it is simple and practical, this method is the most preferred method in practice. By using the determined reduction coefficient (DF), the allowable bearing capacity of the rock mass is calculated with the help of the following formula:

$$qa = \sigma_c - (\sigma_c \times Df) \tag{3}$$

qa: Allowable bearing capacity (MPa)
σci: Uniaxial compressive strength of rock material (MPa)
Df: Reduction coefficient (%)

In Table 4, the input parameters used and the bearing capacity values calculated according to the equations proposed by different researchers are given. In this case, it is seen that the bearing capacity values of the tuffs forming the foundation's soil vary between 1.6–2.0 MPa.

**Table 4.** Bearing capacity values calculated according to the equations proposed by different researchers.

| | Input Parameters | Uniaxial Compressive Strength ($\sigma_c$) | Rock Quality Designation (% RQD) |
|---|---|---|---|
| | Value | 20 MPa | 30 |
| Allowable bearing capacity (MPa) | Bishnoi (1968) | 2.0 MPa | |
| | Peck vd. (1974) | 1.6 MPa | |

The floor area of each cell belonging to the gabion walls is 2 m². In this case, four gabions will be built along each 2 m line at the base and the total floor area will be 8 m². The gabion wall load in this area is 50 tons on average. The contact pressure was calculated as 0.62 MPa. In this case, no problems are expected in terms of bearing capacity.

Tuff samples from the research pits and dasit samples from the crops were taken. Density, unit weight, point load, and uniaxial compressive strength tests on the rock samples taken were carried out in accordance with ASTM and ISRM [39–42] standards. The values of the index and strength properties of the rock material are summarized in Table 5.

**Table 5.** The values of the index and strength properties of the rock material.

| Age | Lithology | Density (gr/cm³) | Unit Weight (kN/m³) | Uniaxial Compressive Strength (MPa) | Point Load (MPa) |
|---|---|---|---|---|---|
| Middle Miocene | Dacite | 2.45–2.56 | 24.0–25.11 | 118 | 5.9–7.86 |
| Middle Miocene | Tuff | 2.14–2.30 | 21.1–22.56 | 20 | 1.1–1.3 |

Within the scope of this study, the average number of discontinuities per meter was determined by carrying out a section study from the outcropping of tuff on which the gabions will be built. An average of 30 discontinuities per meter was determined in the section study, and the following equation was used to calculate RQD.

$$\text{RQD}(\%) = 100e^{-0.1\alpha}(0.1\alpha + 1) \tag{4}$$

$\alpha$: Discontinuities frequency (the number of discontinuities per meter)

### 4.2.2. Inside Stability Failure

Due to insufficient reinforcement-soil friction in internal stability, the cases of stripping of the embankment from the soil are examined [43]. In order to prevent the formation of internal failure, the maximum tensile forces in the reinforcement, the location of these forces on the critical slip surface, and the stresses of the reinforcements against tensile and shear must be determined in the dimension and design phase. The most important issue for the internal loading situation that should be taken into account in reinforced earth walls is the earth pressure of the reinforced soil and the surcharge loads to be located on the upper side of the reinforced zone.

There will be no lateral load on the gabion wall to be constructed within the scope of this study. In addition, there is no surcharge load. Therefore, an internal stability failure due to loads is not expected.

### *4.3. Explicit Analysis*

Within the scope of this study, the structural behavior of the gabion wall, which is designed at a certain height and width as a result of rockfall, has been examined. Structural behavior was determined by simulation based on the finite element model. Within the scope of this study, the ANSYS Workbench [44] was used. ANSYS Workbench, which includes many analysis systems, can combine many engineering studies in one system. The process steps, such as the system to be created for this study, the method of creation, the analysis steps, and the examination of the results, are summarized below.

Explicit analyses are used to simulate dynamic effects occurring in very short periods of time. For this reason, the "Explicit Dynamics" analysis type in ANSYS Workbench was chosen to examine the rockfall effect. By following the seven steps in the analysis system, the desired analysis can be performed and results can be obtained. In step 1, there is the analysis type. No action is required in this section. It is used to avoid confusion that may occur when combined with other analysis types. Material types and properties are determined from the "Engineering Data" tab in step 2. The material properties of the falling rock and the gabion wall system hit by the rock were determined. It is thought that the crushed stone (100–150 mm) inside the gabion wall and the reinforcement wires (3–5 mm) outside the gabion wall will significantly extend the analysis time. For this reason, a macro approach was used in this study. In future studies, it may be more appropriate to make analyses based on micro-analysis and compare the results. In the next step, the geometry is defined. For this, ANSYS Workbench offers two different drawing program plugins (DesignModeler and SpaceClaim). In addition, geometries prepared in other engineering and drawing programs can also be transferred to this program. For this example, gabion

wall and rock sphere geometry prepared on AutoCAD were imported into the program using the drawing file with the extension ".igs" (Figure 16). In the third step ("Model" step), assignment of material properties, meshing, loading conditions, support conditions, etc., parameters are set and analysis settings are made. In this step, the Explicit Dynamics-Mechanical tab is opened. Necessary assignments and adjustments are made on this screen (Figure 17a). Material assignments are performed by selecting the modeled geometries. The "Stiffness behavior" property of the rock has been determined as rigid (Figure 17b). The contact of the objects is defined in the "Body Interactions" section. Therefore, the "Contact" section in the "Connections" menu in the "outline" window should be deleted (Figure 18a). Since it would be sufficient to perform the meshing at default values, parameters were not changed. However, if a more frequent meshing is wanted, the wanted value can be entered in the "Element Size" section (Figure 18b). The impact of the rock against the gabion wall at an angle of 10 degrees and a speed of 30 m/s is modeled (Figure 19a). Explicit analyses are simulating events that occur in very short times, so the simulated times in the analysis are taken as very small in order to reduce the workload. For this reason, the modeled time in the analysis was determined as 0.02 s. This time may be extended within the scope of future studies (Figure 19b). Calculation of the wanted result value can be performed in the solution section of the Outline menu. For this study, the total displacement, equivalent stress, and velocity of the stone were calculated (Figure 20a). The change for the falling rock in a period of 0.02 s is seen on the graph. There is no need to extend the analysis simulation time as the speed drops to 10% of the initial level (Figure 20b). As a result of the analysis, the maximum displacement value was obtained as 13,655 cm (Figure 21). The stresses on the wall are given in Figure 22. Crash zone damage is given in Figure 23. Upon examining Figure 23, it is clearly seen that the damage remained in a limited area of the gabion wall, its structural integrity was not compromised, and the wall did not undergo a collective collapse, thus fulfilling its function. It can be seen from Figures 22 and 23 that the impact of the rock is absorbed by the entirety of the wall. As stated, analysis is stopped when the rock's velocity drops to 10% of its initial speed. This results in approximately 99% of the rock's energy being transferred to the gabion wall. Except for minor damage on the contact face, the wall remained intact. The highest stress levels were observed on four blocks that were hit by the rock and the highest equivalent stress was below 30 MPA. As expected, the purpose of such structural measures is for the gabion wall to perform its function by incurring damage and to protect the settlements located behind the wall. Therefore, the designed gabion wall has demonstrated the expected behavior by sustaining limited damage.

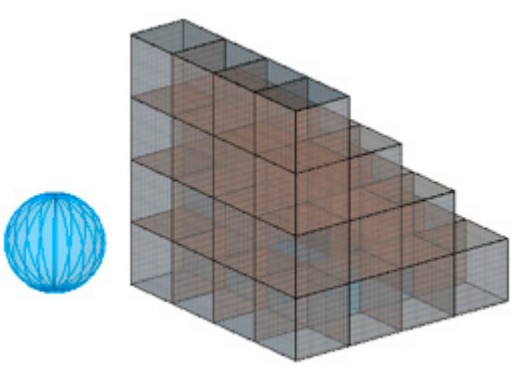
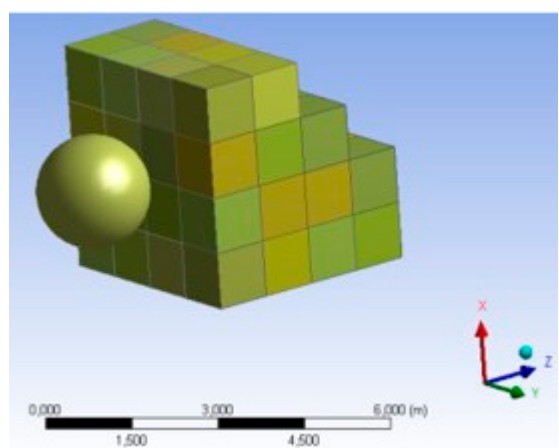

**Figure 16.** Screen interface of the modeling phase of Explicit Dynamics analysis.

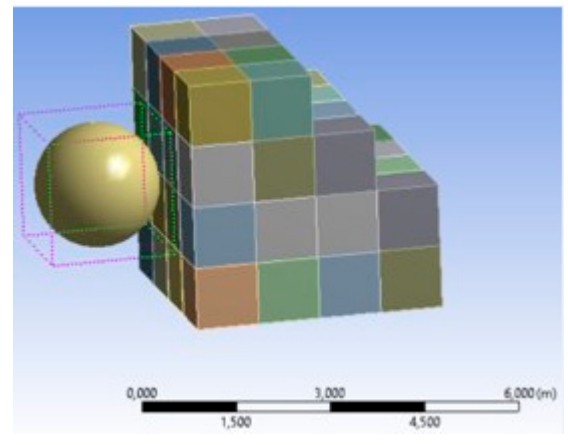

(**a**) Explicit Dynamics-Mechanical tab

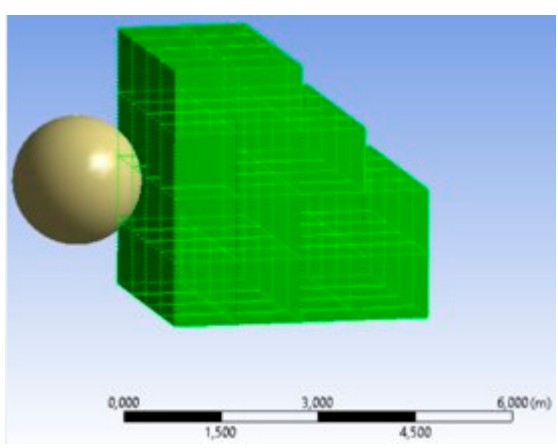

(**b**) Assigning material properties

**Figure 17.** Screen interface of Explicit Dynamics-Mechanical tab and assigning material properties.

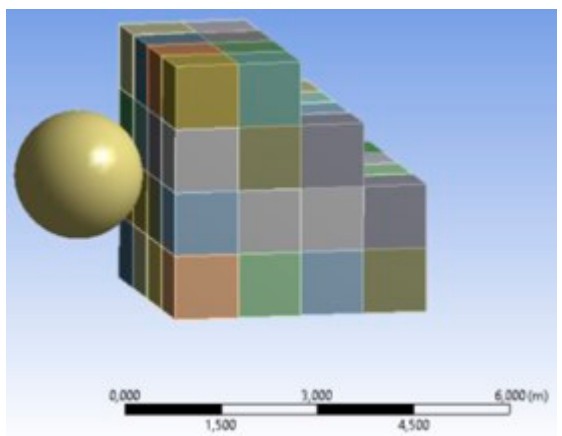

(**a**) Explicit Dynamics-Connections/Contact tab

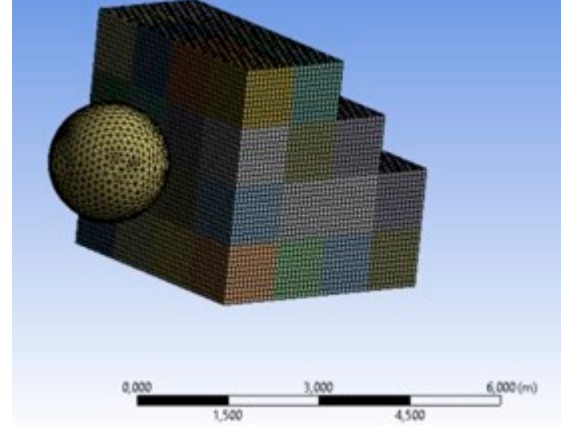

(**b**) Explicit Dynamics-Meshing tab

**Figure 18.** Screen interface of Explicit Dynamics-Connections/Contact tab and the Explicit Dynamics-Meshing tab.

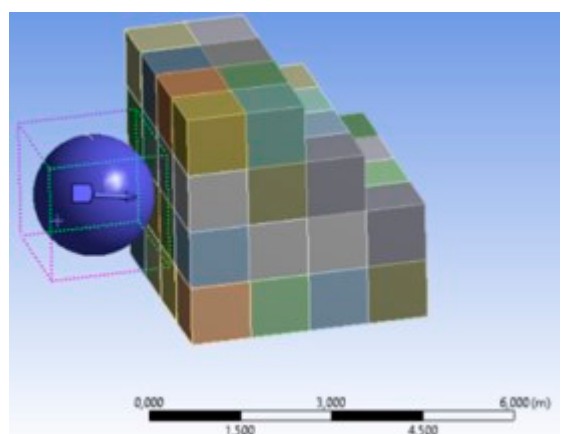

(**a**) Modeling the fall angle and velocity of the rock

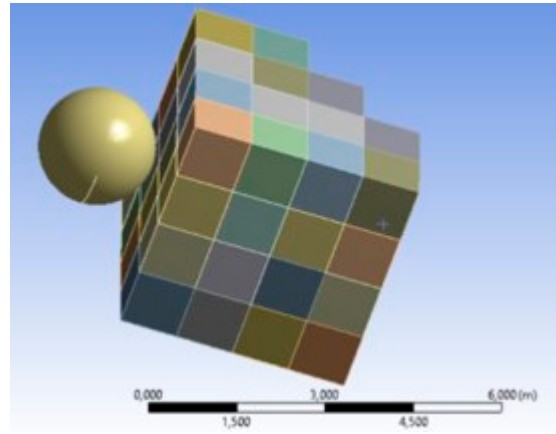

(**b**) Determining the analysis time

**Figure 19.** Screen interface for modeling the fall angle and velocity of the rock and determining the analysis time.

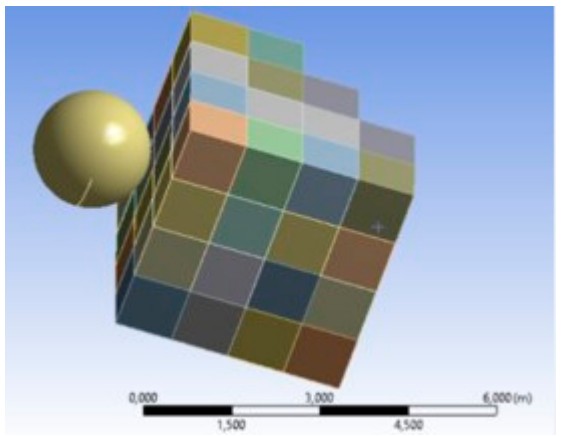

(**a**) The data to be obtained after the analyses

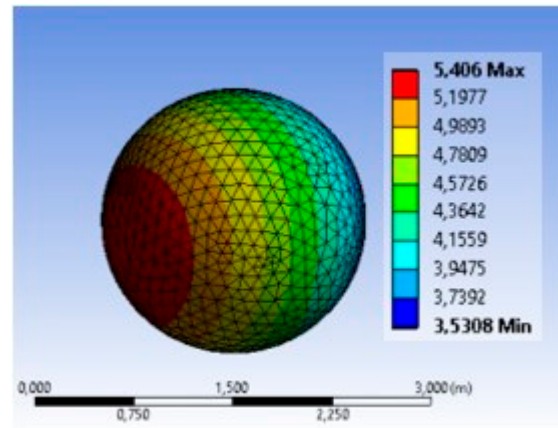

(**b**) The change for the falling rock in 0.02 s

**Figure 20.** Screen interface of the data to be obtained after the analyses and display interface of the change for the falling rock in 0.02 s.

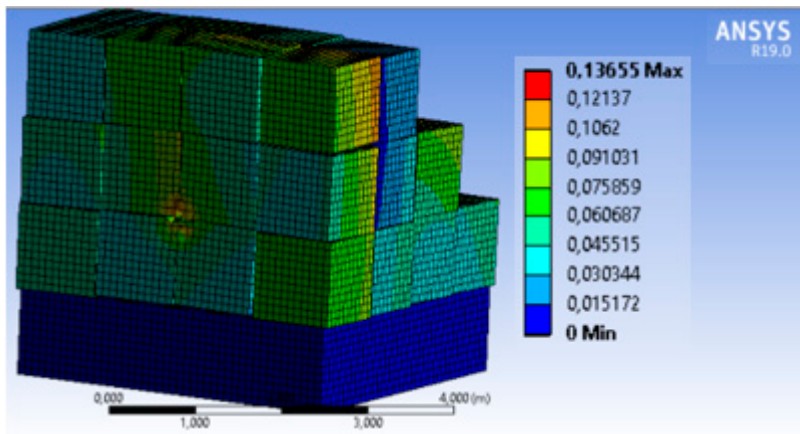

**Figure 21.** Screen interface of the maximum displacement obtained as a result of the analyses.

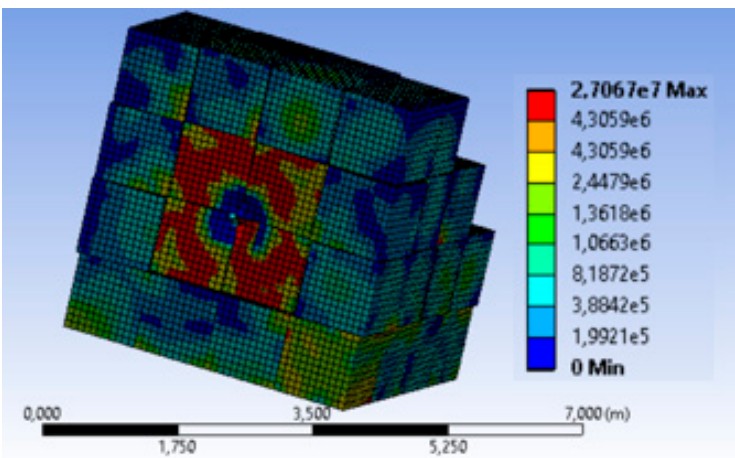

**Figure 22.** As a result of the analyses, the screen interface of the stresses.

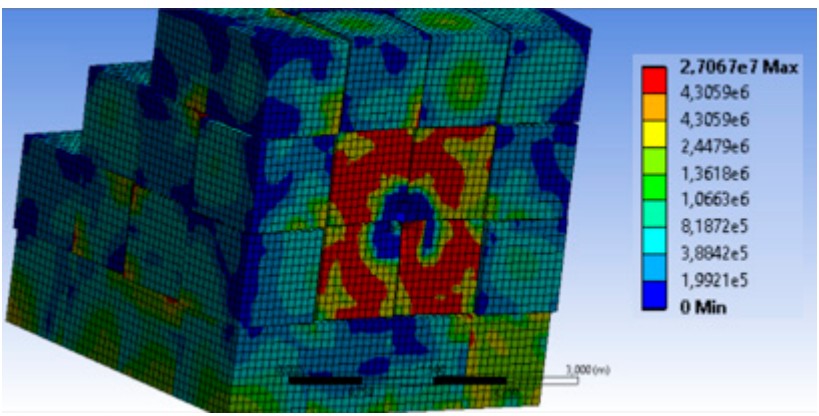

**Figure 23.** Screen interface of the crash damages obtained as a result of the analyses.

## 5. Conclusions

Within the scope of this study, the cases of rock falls that occurred on a slope in the east-northeast part of Çortak Village in Selendi District of Manisa Province were evaluated. The routes, jump heights, and energies of the blocks that will move from the source rock were determined by 3D analysis. According to the numerical data obtained from the 3D maps, different applications were evaluated to minimize the risk of rockfall. All the data obtained were compiled as a whole and the most suitable reclamation method(s) for the study area were determined. The results are summarized below:

There are rhyolitic and dacidic rock masses, especially at higher elevations of more than 800 m.

Severe erosion is observed in and around the study area due to lithological and geomorphological features. The high drainage density, the differently visible gully areas, and the sharp ridges separating them indicate that the erosion is severe. This process, which especially affects rocks with low strength, also increases rockfall events in rock masses located at higher elevations.

The study site starts from Çortak Village (Selendi, Manisa) and extends to the northwest for approximately 700–800 m. The slope increases towards the north and reaches 40–50 degrees at the upper elevations. At lower elevations, the slope decreases to about 20 degrees. Generally, the slope tendency is southwest. The rock blocks that broke off from the source rock masses in NW-SE trending moved in the southwest direction and reached a level threatening the village.

Many blocks that break off from the source rock complete their movement at the point where the energy ends, and they vary in size from 10 cm to 2 m to pose a risk on the slope. It is seen that some blocks up to 2 m reach the areas very close to the village by moving across approximately 400–500 m from the source rock.

The rockfall risk weighted score for the slope was calculated as 49.90. This value shows that the rock slope in the study area is classified as "medium risk slope (risk class III)".

In the 3D rockfall analysis applied, considering the worst case, blocks with a mass of 10 tons were dropped along a line from the highest point of the source rock. Fifty blocks were modeled in each region.

When the analyses were evaluated, it was observed that the jump heights for a 10-ton block exceeded 2 m on the steep cliffs at the top, 1 m on average along the slope, and only turned into a rolling shape at the lowest elevations (areas where the gabion design was planned). However, jump heights of 3 m were recorded in some parts of the lower elevations. At these points, the energy amounts vary between 4000–8000 kJ for a 10-ton block, and the average speed reaches 30 m/s.

Gabion walls are generally designed to eliminate the risks that may arise from large blocks with low jump height but very high energy. This is also the case in the study area. In addition, the fact that there is a buffer (flat) zone between the places and the base of the slope shows that structures such as gabions can be applied on the slope.

Gabion walls were modeled along five lines to the most suitable locations. For this case, the rockfall analyses were repeated. When the 3D analyses made after the design of the gabion wall are evaluated, it is seen that the design is safe.

Bearing capacity analyses were made in the area where the gabion wall will be built. It is seen that the bearing capacity values of the tuffs forming the foundation soil vary between 1.6–2.0 MPa. Considering the 0.62 MPa contact pressure, no problem is expected in terms of bearing capacity.

Within the scope of this study, the structural behavior of the gabion wall, which is designed at a certain height and width as a result of rockfalls, has been examined. Structural behavior was determined by simulation based on the finite element model. In the analyses, the ANSYS Workbench program was used. The impact of the rock against the gabion wall at an angle of 10 degrees and a speed of 30 m/s is modeled. The "Explicit Dynamics" analysis type in ANSYS Workbench was chosen to examine the rockfall effect. As a result of the analysis, the maximum displacement value was obtained as 13,655 cm. Also, the stresses distribution on the wall are obtained.

The results obtained revealed that it is extremely important to test the reliability of the proposed method by numerical analyses in addition to the rockfall analysis.

**Author Contributions:** Investigation, Z.A. and O.Ş.K.; resources, Z.A. and O.Ş.K.; writing—original draft preparation, Z.A. and O.Ş.K.; writing—review and editing, Z.A. and O.Ş.K. All authors have read and agreed to the published version of the manuscript.

**Funding:** This research received no external funding.

**Data Availability Statement:** The original contributions presented in the study are included in the article, further inquiries can be directed to the corresponding author.

**Conflicts of Interest:** All authors declare that the research was conducted in the absence of any commercial or financial relationships that could be construed as a potential conflict of interest.

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
