# Peer review of "Evaluation of the Performance of Gabion Walls as a High-Energy Rockfall Protection System Using 3D Numerical Analysis: A Case Study"

_applsci, doi:10.3390/app14062360_

Round 1

Reviewer 1 Report

Comments and Suggestions for Authors

This paper evaluates the performance of gabion walls as a high-energy rockfall protection system using 3d numerical analysis based on a case study. The research is interesting. The following issues require to be addressed.

1) It seems that the terrain analysis and the rockfall analysis are independent. More relationship should be addressed for these two parts.

2) The rockall analysis is too simple, more parametric analyses are required to discuss different energy, heights, directions, shape, etc.

3) The technical disscussion is too shallow.  It requires deeper analysis on the impact energy transfer and dissipation.

4) The simulation basics needs to be further addressed. The software, settings, governing equations, basic assumptions.

5) The simulation use spheric impact objects, what are the influence of different shapes? What if multiple rockfalls? What is the influence of the wall length? These should be discussed further.

Reviewer 2 Report

Comments and Suggestions for Authors

Evaluation of performance of gabion walls as a high-energy rockfall protection system using 3d numerical analysis: a case study

This article examines the instances of rockfalls on a slope within Selendi District, Manisa Province. Through field investigations, various strategies to mitigate the risk of rockfalls were explored. Drone surveys provided a basis for evaluating previous assessments and for generating digital topographic maps at a 1/1000 scale. Utilizing these maps, a 3D solid representation of the project area was constructed. Field inspections were carried out to pinpoint the origins of potentially hazardous rocks and detached blocks. The analysis of 3D data facilitated the prediction of movement trajectories, bounce heights, and kinetic energies of the dislodged rocks. This information was instrumental in selecting the optimal countermeasures for rockfall mitigation. The research provides valuable insights and design mitigate the risk of rockfalls. However, several areas require clarification and improvement. I recommend the following revisions:

1. Introduction: The structure of the entire introduction needs reorganizing, including enhancing the overly simplistic third paragraph and simplifying the excessively lengthy fourth paragraph.

2. Introduction: Please pay attention to details such as the order of references.

3. Introduction: The fourth paragraph needs organizing, rather than simply listing previous studies. For instance, presenting a table or flow chart to analyze the different important factors considered in rockfall events could be more effective. The following publication is so useful for this kind of works.

[1] Cao B, Ghavidelnia N, Speck O, Eberl C. Flow charts as a method to transfer self-sealing from plant models into programmable materials and related challenges. Program Mater 2023;1:e12. https://doi.org/10.1017/pma.2023.11.

Moreover, the scope of this section, which encompasses a wide range of studies on rockfall events is too broad for this article; such as PE and EPE in Ref. [42], it does not cover these materials in this study. Therefore, is it possible to focus solely on the role of gabion walls in rockfall events in this section?

4. Introduction: Please standardize the usage of "3D" and "three-dimensional."

5. Introduction (Line 177 to Line 178): “Drone flights were utilized to assess prior research and to acquire digital topographic maps at a 1/1000 scale.” More details need to be provided about this section. Relevant research methods should be consolidated into a new section, such as Part Three: Research Methods.

6. Results: Similar to the previous suggestion, too much information on research methods is included in this section; therefore, the methods section needs to be consolidated into a new chapter.

7. Results (Line 230): Minor revision: there is an extra period.

8. Results (Figure 6): Minor revision: In this figure, "RP 1" and "RP2" should be corrected to "RP-1" and "RP-2".

9. Results (Figure 6): Please provide the heights corresponding to RP1 through RP5, as different heights may have a significant impact.

10. Results (Figure 13): From the diagram, it appears that the highest bounce could be around 2 meters. However, if the gabion wall is set to 4 meters, the cost would increase. Although ditches cannot stop rockfalls of about 2 meters, their construction cost is lower than that of gabion walls. Therefore, is it possible to provide a suitable range for gabion wall construction based on parameters such as the slope, speed of movement, etc.?

11. Results (Figures): The clarity of all images in this study is very low; it is necessary to significantly improve the resolution of the figures.

12. Results (Figure 14): Unit is missing in Figure 14.

13. Results (Line 394): The full name of RQD, which stands for Rock-quality designation, needs to be provided.

14. Results (Line 430 and 431): Minor revision: "2 m2" "8 m2"

15. Results: Further in-depth analysis of the results in Figure 22 is required.

16. Conclusions:Please polish the language; the phrase "Within the scope of the study" has been repeated too many times.

Round 2

Reviewer 1 Report

Comments and Suggestions for Authors

The two issues are not fully addressed in the revised manuscript.

1) The rockfall analysis is too simple, more parametric analysis are required different energy, height, direction and shape...

2) The technical discussion is too shallow. It requires deeper analysis of the impact energy transfer and dissipation.

Reviewer 2 Report

Comments and Suggestions for Authors

All the comments are well addressed. The manuscript is recommended to be accepted.

Author Response

Thank you for all your valuable comments.

Round 3

Reviewer 1 Report

Comments and Suggestions for Authors

The revision can be accepted.